# Taxonomic Delimitation of the Monostromatic Green Algal Genera *Monostroma* Thuret 1854 and *Gayralia* Vinogradova 1969 (Ulotrichales, Chlorophyta)

**Jianjun Cui** [1], **Chunli Chen** [1], **Huaqiang Tan** [2], **Yongjian Huang** [1], **Xinyi Chen** [1], **Rong Xin** [1], **Jinlin Liu** [3], **Bowen Huang** [1] and **Enyi Xie** [1,*]

1  Fishery College, Guangdong Ocean University, Zhanjiang 524088, China
2  Southern Marine Science and Engineering Guangdong Laboratory (Zhanjiang), Zhanjiang 524006, China
3  College of Marine Ecology and Environment, Shanghai Ocean University, Shanghai 201306, China
*  Correspondence: xieenyi@163.com

**Abstract:** The genera *Monostroma* and *Gayralia* belong to the order of monostromatic green algae; however, their taxonomic delimitation remains controversial at the genus level. This study attempts to address this issue through the combined analysis of the morphology and nuclear-encoded Internal Transcribed Spacer region sequences of monostromatic green algal samples collected in the South China Sea. Our phylogenetic data revealed that the monostromatic specimens were separated into the *M. nitidum* clade, *G. brasiliensis* clade, and a single *Monostroma* sp. clade, and that the inter-genera genetic distance between the *Monostroma* and *Gayralia* genera was lower than that observed within the *Monostroma* genus. All the specimens presented similar morphology in their single cell-layered thallus, with irregularly arranged cells, rounded cell corners, a parietal chloroplast, and predominantly one (>90%) pyrenoid. Their most obvious morphological difference was in thallus thickness and size. Moreover, the monostromatic specimens of the *M. nitidum* clade corresponded to the morphological description of the *M. nitidum*-type specimens. The genus *Monostroma* was erected earlier than the genus *Gayralia*. Therefore, we propose to assign the genus *Gayralia* to *Monostroma* based on the morphological and phylogenetic analysis and genetic distance data presented here.

**Keywords:** *Gayralia*; Internal Transcribed Spacer sequence; *Monostroma*; morphology; taxonomy





## 1. Introduction

Monostromatic green algae with fronds consisting only of horizontally arranged single-cell layers are widely distributed from temperate to tropical seas worldwide [1]. *Monostroma* Thuret 1854, *Gayralia* Vinogradova 1969, and *Protomonostroma* Vinogradova 1969 are among the main monostromatic green algal genera [2–5]. *Monostroma* is cosmopolitan and comprises 55 species, of which only 32 are currently taxonomically confirmed [6], while for the *Gayralia* and *Protomonostroma* genera, only two species per genus have been confirmed, namely *G. brasiliensis* and *G. oxysperma* for the former, and *P. undulatum* and *P. rosulatum* for the latter [6]. A number of monostromatic green algal species are attracting global attention due to their economic importance, mainly in the food and cosmetic industries [7–10]. In addition, chemicals with antiviral and anticoagulant properties were recently isolated from some species of the *Monostroma* and *Gayralia* genera [11,12].

*Monostroma* and *Gayralia* are the focus of multidisciplinary research involving different fields, such as taxonomy, biology, phylogeny, and biogeography [4,5,10,13,14]. As such, correct species identification and classification are essential to enable further research. Currently, the taxonomy of the *Monostroma* and *Gayralia* genera is complex, as there are several inconsistencies among the different classification tools, and experts disagree on the biological features considered relevant for the separation of taxa. Phenotypic variation in monostromatic green algae is well documented. *Monostroma*, which is characterized by a

blade-shaped thallus consisting of one layer of cells, was erected by Thuret in 1854 [15], and it was later lectotypified with *M. oxyspermum* [16]. Species belonging to this genus are classically defined based on morphological characteristics, such as the size and shape of the cells and the thickness of thalli [17]. Culture studies have been conducted in at least some of the taxa, resulting in taxonomical revisions. For instance, Kornmann (1964) and Bliding (1968) proposed to remove the asexual *M. undulatum* and *M. oxyspermum* from the *Monostroma* genus [18,19]. Subsequently, the Gayraliaceae family—comprising two monotypic genera, *Protomonostroma* and *Gayralia*—was erected by Vinogradova (1969) to accommodate the asexual members, respectively [3]. *Ulvaria oxysperma* and *M. oxyspermum* have been synonymized with *G. oxysperma* based on thallus ontogeny and flagellar ultrastructural features [20]. A preliminary molecular analysis of Internal Transcribed Spacer (ITS) sequences revealed that *M. nitidum* and *M. latissimum*, the typical species of *Monostroma*, should be transferred to *Gayralia* [5]. In contrast, Bast (2011) concluded that *Gayralia* should be abolished and its species should be transferred back to *Monostroma*, because no obvious taxonomic distinction was detected between the two genera [4]. Furthermore, the Gayraliaceae family is posterior to the Monostromataceae, Kunieda ex Suneson (1947) and, based on the priority principle, the retention of the latter with the original *Monostroma* genus would be appropriate [3,4,16]. Thus, considering these contradictions and ambiguities, the current taxonomic status of these two genera warrants clarification.

Morphological differences across specimen types were confirmed as being reliable in taxonomically classifying macroalgae [10,21,22]. To address the taxonomic issues mentioned above, a morphological analysis of the specimen type and targeted monostromatic samples collected from the South China Sea, was conducted. The identification of monostromatic algae based solely upon morphological features is extremely challenging, and studies based on life cycle completion—although they often aid in the identification—are time consuming and difficult [3,18,19]. Molecular phylogenetic analyses compensate for the shortage of morphological studies, and they provide an accurate identification of ITS sequences, which are now available for a large group of marine green algae and have a high degree of variance, even between very closely related algal taxa [23]. Moreover, morphological characteristics, combined with sequence analysis, have also been used to resolve identification issues at the species level [10,21,22,24–30]. Therefore, in the present study, nuclear-encoded rDNA ITS sequences were obtained from monostromatic green algae collected in the South China Sea and from previously identified *Monostroma* and *Gayralia* species to characterize sequence divergence between these two genera.

## 2. Materials and Methods

### 2.1. Monostromatic Green Algal Collection

Attached monostromatic algal samples were collected from the following areas of the South China Sea coast in the Guangdong province: Zhanjiang, May 2020; Maoming, February 2020; Yangjiang, November 2019 and February 2020; Zhuhai, March 2021; and Shantou, March 2021 (Figure 1). The samples (Figure 1B–F) were placed in a cooler on ice and were brought to the laboratory. Before identification, sediments and contaminants were removed using filtered seawater and a soft brush. Detailed sample information is listed in Table 1.

### 2.2. Morphological Examination

Several intact monostromatic green algae from each sampling station were selected for morphological assessment, and their macroscopic features—including thallus type, size (length × perpendicular width), and color—were recorded. In addition, surface view and transverse section microphotographs of the cells were obtained under a microscope (Olympus CX33, Tokyo, Japan), and they were used to determine the microscopic cellular features, including cell size, shape and arrangement, chloroplast shape, position, and pyrenoid number.

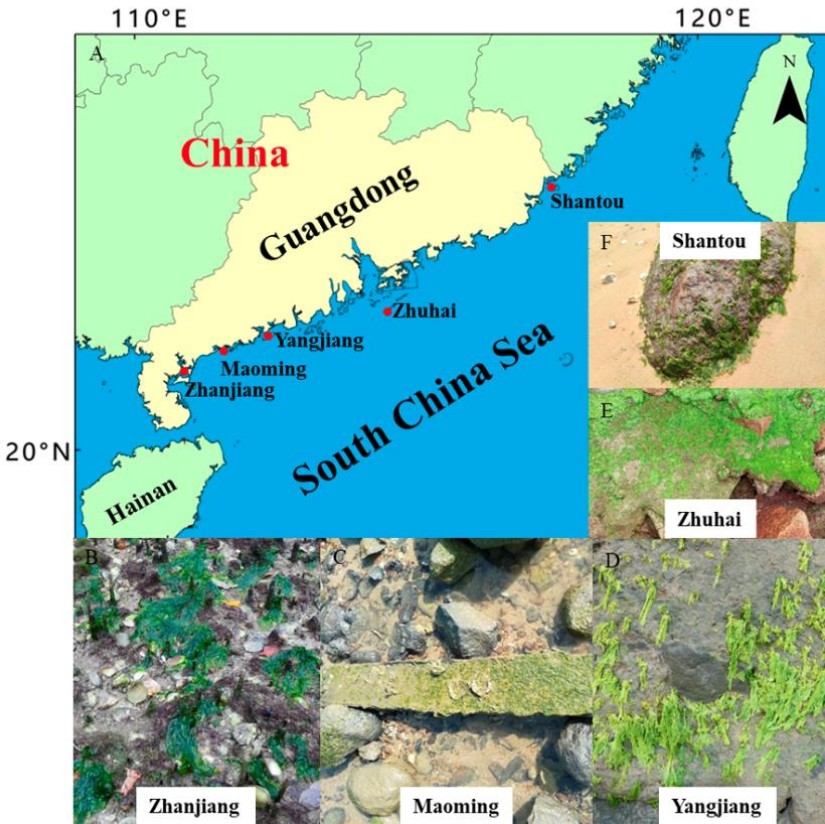

**Figure 1.** Sampling sites where monostromatic green algae were collected from the South China Sea. (**A**) Map of the sampling stations; (**B–F**) images depicting the monostromatic samples sites.

**Table 1.** Sample information concerning the monostromatic species used in this study.

| Taxon | Strain Code | Collection Locality | Collection Date | Accession No. of ITS | Reference |
|---|---|---|---|---|---|
| Monostromatic species | ZJ01 | Dalang, Naozhou Island, Zhanjiang, China | 05-May-2020 | OP151123 | This study |
| | ZJ02 | Dalang, Naozhou Island, Zhanjiang, China | 05-May-2020 | OP151124 | This study |
| | ZJ03 | Techeng Island, Zhanjiang, China | 15-May-2020 | OP151125 | This study |
| | ZJ04 | Techeng Island, Zhanjiang, China | 15-May-2020 | OP151126 | This study |
| | MM01 | Xiaofangji Island, Maoming, China | 19-Feb-2020 | OP151105 | This study |
| | MM02 | Xiaofangji Island, Maoming, China | 19-Feb-2020 | OP151106 | This study |
| | MM03 | Xiaofangji Island, Maoming, China | 19-Feb-2020 | OP151107 | This study |
| | YJ01 | Yinglu, Hailing Island, Yangjiang, China | 10-Feb-2020 | OP151102 | This study |
| | YJ02 | Yinglu, Hailing Island, Yangjiang, China | 10-Feb-2020 | OP151103 | This study |
| | YJ03 | Yinglu, Hailing Island, Yangjiang, China | 10-Feb-2020 | OP151104 | This study |
| | YJ04 | Mali, Hailing Island, Yangjiang, China | 01-Nov-2019 | OP151127 | This study |
| | YJ05 | Mali, Hailing Island, Yangjiang, China | 01-Nov-2019 | OP151128 | This study |
| | ZH01 | Miaowan Island, Zhuhai, China | 18-Mar-2021 | OP151108 | This study |
| | ST01 | Nanao Island, Shaotou, China | 23-Mar-2021 | OP151129 | This study |
| | ST02 | Nanao Island, Shaotou, China | 23-Mar-2021 | OP151130 | This study |
| | ST03 | Nanao Island, Shaotou, China | 23-Mar-2021 | OP151131 | This study |
| *Monostroma nitidum* | / | Bailongwei, Fangcheng, Guangxi, China | / | AF415170 | [31] |
| | / | / | / | AY026917 | [5] |
| *Monostroma kuroshiense* | / | Sakurajima, Kagoshima Pref, Japan | Mar/Apr-2009 | GU062561 | [4] |
| *Monostroma arcticum* | / | Shimiao, Dalian, China | / | AF415171 | [31] |
| | / | Donegal, Ireland | Mar/Apr-2009 | GU062560 | [4] |
| *Monostroma grevillei* | / | Xiaoping Island, Dalian, China | / | AF428051 | [31] |
| | / | Atlantic | / | AF499456 | [32] |
| | / | Baltic | / | AF499457 | [32] |

**Table 1.** *Cont.*

| Taxon | Strain Code | Collection Locality | Collection Date | Accession No. of ITS | Reference |
|---|---|---|---|---|---|
| | / | Florianópolis, Santa Catarina, Brazil | 01-Jul-2006 | KC143761 | [5] |
| | / | Guaratuba Bay, Paraná, Brazil | 11-Aug-2006 | KC143762 | [5] |
| *Gayralia* | / | Piúna Beach, Piúna, Espírito Santo, Brazil | 12-Sep-2007 | KC143766 | [5] |
| *brasiliensis* | / | Caravelas, Bahia, Brazil | 26-Jan-2008 | KC143768 | [5] |
| | / | Itapissuma, Pernambuco, Brazil | 09-May-2007 | KC143770 | [5] |
| | / | / | / | AY016306 | [5] |
| *Gayralia* | / | Antonina Bay, Paraná, Brazil | 09-Jul-2006 | KC143758 | [5] |
| *oxysperma* | / | Lagoinha Beach, São Paulo, Brazil | 07-Aug-2005 | KC143759 | [5] |
| | / | Laguna, Santa Catarina, Brazil | 23-Nov-2010 | KC143760 | [5] |
| *Ulva prolifera* | / | Sekiguchi R., Yamada, Iwate, Japan | 02-May-2005 | AB298316 | [33] |
| *Ulva* sp.2 | / | Yoshino River, Tokushima, Japan | 18-Jul-2000 | AB298457 | [33] |

"/" stands for data not available.

### 2.3. DNA Extraction

Five monostromatic samples from Yangjiang (strain codes: YJ01, YJ02, YJ03, YJ04, and YJ05), four from Zhanjiang (strain codes: ZJ01, ZJ02, ZJ03, and ZJ04), three from Maoming (strain codes: MM01, MM02, and MM03) and Shantou (strain codes: ST01, ST02 and ST03), and one from Zhuhai (strain code: ZH01), which were randomly selected during the morphological examination, were prepared for DNA extraction. All samples were crushed into a fine powder in liquid nitrogen after being dried under vacuum conditions. Total DNA was extracted from each dried sample using a DNEasy Plant Mini kit (Qiagen, Valencia, CA, USA) following the manufacturer's instructions. The quality of the extracted DNA was confirmed by electrophoresis on 1% agarose gel in TAE buffer containing ethidium bromide (EtBr) under ultraviolet (UV) light. The extracted DNA was stored at 4 °C until further use.

### 2.4. PCR Amplification

Polymerase chain reaction amplifications were performed in a total volume of 20 μL containing 2 μL of template DNA, 2 μL of each primer, and 10 μL 2 × San *Taq* PCR mix (Sangon, Shanghai, China). The primers used to amplify all the ITS sequences were ITS 1 and ITS 4 [23], and the reaction cycles were: initial denaturation at 94 °C for 5 min, followed by 28 cycles at 94 °C for 30 s, 55 °C for 30 s, 72 °C for 60 s, and 10 min at 72 °C as the final extension step. PCR products were checked on 1% TAE agarose gels stained with ethidium bromide, and were sequenced by Shanghai Sangon Corp. (Sangon, Shanghai, China).

### 2.5. Phylogenetic Analysis

Sequences were aligned with published data (Table 1) using ClustalX [34] and edited in BioEdit [35]. For comparative analyses, sequences (AB298316 and AB298457) downloaded from Genbank were utilized as an outgroup. Finally, 35 nrITS sequences were used for phylogenetic analyses which, for concatenated nrITS, were performed using the maximum likelihood (ML) and neighbor-joining (NJ) methods in Molecular Evolutionary Genetics Analysis (MEGA) v.7.0.21 [36]. The best-fitting model of nrITS for the ML and NJ analyses was the Tamura-Nei + Gamma distribution (G) + Invariant sites (I) and their robustness was tested by bootstrapping with 1000 replicates. Bayesian inference (BI) analysis was performed using MrBayes v3.1.2 [37]. The best partition strategy and model of sequence evolution were selected based on the Bayesian Information Criterion (BIC). Four chains of Markov chain Monte-Carlo iterations were performed for 1,000,000 generations, keeping one tree every 100 generations. Convergence of the runs was checked visually with Tracer v1.6 [38]. A burn-in of 25% was used to avoid suboptimal trees in the final consensus tree. The pairwise distances of nrITS were calculated based on the Maximum Composite Likelihood model using MEGA v.7.0.21 [36].

## 3. Results

### 3.1. Phylogenetic Analyses

The ITS 1 and ITS 2 regions that included the 5.8S gene were successfully amplified and sequenced from 16 target samples. Sequences of 546 bp from each of the ZJ01-04, ST01-03, and YJ04-05 samples, and 554 bp from each of the YJ01-03, MM01-03, and ZH01 samples, were used for the phylogenetic analyses. The phylogenetic trees obtained from the ML, NJ, and BI analyses of the nrITS and concatenated data revealed that the monostromatic green algae fell into three distinct clades (Figure 2). Specifically, nine samples (ZJ01-04, ST01-03, and YJ04-05), together with the identified *M. nitidum* from China, were resolved in the *M. nitidum* clade (99% in ML and NJ, 0.99 in BI); three samples (YJ01-03), together with the identified *G. brasiliensis* from Brazil, were resolved in the *G. brasiliensis* clade (100% in ML and NJ, 1 in BI); and four additional samples (MM01-03, ZH01) were included in a unique *Monostroma* sp. clade (100% in ML and NJ, 1 in BI). The intraspecific genetic distance of monostromatic green algae in the *Monostroma* and *Gayralia* genera was less than 0.6% (Figure 3), while their interspecific genetic distance was undulatory. Surprisingly, the inter-genera genetic distance (between the *M. nitidum* and *G. brasiliensis* clades = 0.058–0.062; between the *M. nitidum* and *G. oxysperma* clades = 0.208–0.211) was lower than that observed within the *Monostroma* genus (between the *M. nitidum* clade and the panmictic *Monostroma* sp. clade = 0.291–0.313) (Figure 3).

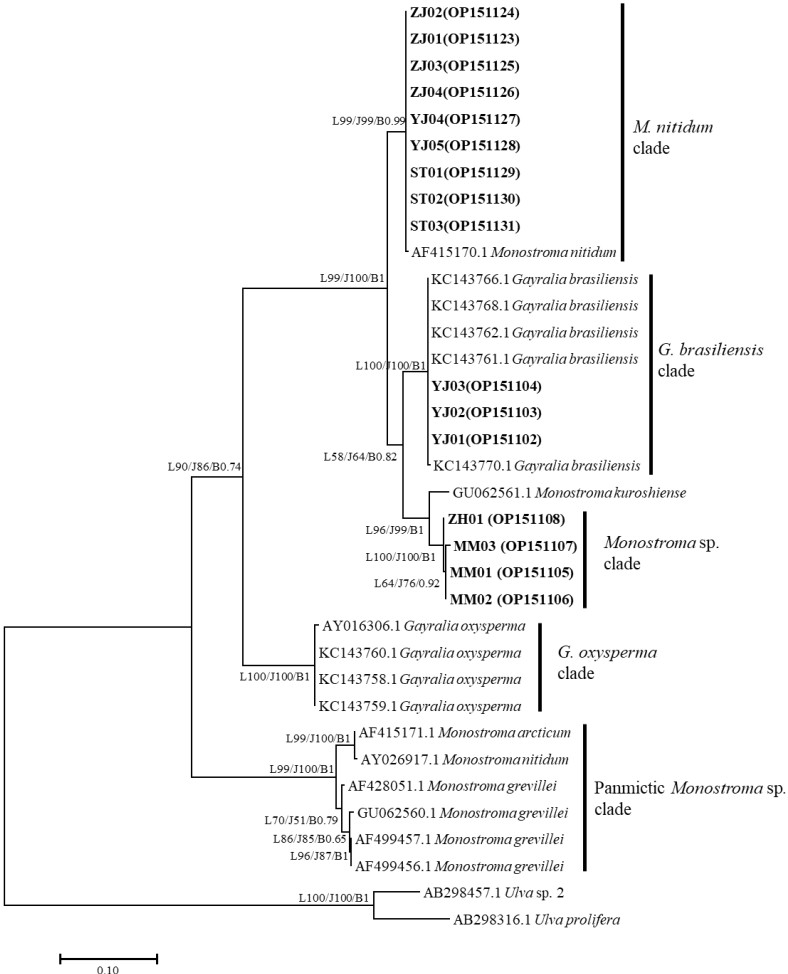

**Figure 2.** Phylogenetic tree inferred from the ITS sequences of the monostromatic green algae collected from the South China Sea and those downloaded from GenBank. The specimens from this study are shown in bold. The numbers on the branches indicate bootstrap values from ML (L . . . , left), NJ (J . . . , middle), and Bayesian inference posterior probabilities (B . . . , right). Bootstrap values (>50%) and Bayesian inference posterior probabilities (>0.50) are indicated.

| | | | | | | | | | | | | | |
|---|---|---|---|---|---|---|---|---|---|---|---|---|---|
| ZJ01-04, ST01-03, YJ04-05 | 0.000 | | | | | | | | | | | | |
| AF415170 *M. nitidum* | 0.002 | 0.000 | | | | | | | | | | | |
| YJ01-03 KC143761 *G. brasiliensis* | 0.058 | 0.060 | 0.000 | | | | | | | | | | |
| KC143770 *G. brasiliensis* | 0.060 | 0.062 | 0.002 | 0.000 | | | | | | | | | |
| MM01-02 | 0.070 | 0.073 | 0.070 | 0.073 | 0.000 | | | | | | | | |
| MM03 | 0.075 | 0.078 | 0.075 | 0.078 | 0.004 | 0.000 | | | | | | | |
| ZH01 | 0.068 | 0.070 | 0.068 | 0.070 | 0.002 | 0.006 | 0.000 | | | | | | |
| GU062561 *M. kuroshiense* | 0.079 | 0.081 | 0.065 | 0.067 | 0.038 | 0.043 | 0.036 | 0.000 | | | | | |
| AY016306 *G. oxysperma* | 0.211 | 0.208 | 0.215 | 0.217 | 0.229 | 0.229 | 0.225 | 0.223 | 0.000 | | | | |
| AF499456 *M. grevillei* | 0.294 | 0.291 | 0.314 | 0.317 | 0.343 | 0.352 | 0.338 | 0.333 | 0.248 | 0.000 | | | |
| AF415171 *M. arcticum* | 0.309 | 0.305 | 0.331 | 0.335 | 0.369 | 0.369 | 0.364 | 0.358 | 0.259 | 0.033 | 0.000 | | |
| AY026917 *M. nitidum* | 0.313 | 0.309 | 0.335 | 0.339 | 0.364 | 0.364 | 0.359 | 0.353 | 0.262 | 0.036 | 0.002 | 0.000 | |
| AB298316 *Ulva prolifera* | 0.495 | 0.490 | 0.496 | 0.496 | 0.482 | 0.487 | 0.487 | 0.492 | 0.486 | 0.491 | 0.499 | 0.500 | 0.000 |

**Figure 3.** ITS sequence-based genetic distances between monostromatic green algal strains collected from the South China Sea and those downloaded from GenBank.

### 3.2. Morphology of Monostromatic Green Algae

Figure 4 shows the morphology of the collected monostromatic green algae. In the target samples, thalli with similar morphology presented a yellowish or light green color, were flat, and had a single-cell layer (Figure 4). In the surface view, cells appeared coupled, irregularly arranged, and triangular or polygonal with three to five rounded corners, partly oval, and paired (Figure 4B,E,H,K,N). A single prominent chloroplast covered most of the outer cell in the surface view and contained predominantly one ($\geq$96%) and occasionally two ($\leq$4%) pyrenoids (Figure 4B,E,H,K,N and Table 2). Cells in transverse sections were circular or quadrangular with rounded corners (Figure 4C,F,I,L,O). However, the thallus shape, cell size, and thallus thickness in the samples presented distinct differences (Figures 4 and 5 and Table 2). Specifically, the thallus of the monostromatic species in the *G. brasiliensis* clade was approximately 3.64 $\pm$ 0.81 cm long and 2.66 $\pm$ 1.41 cm wide, which was larger than that of the *Monostroma* sp. clade (1.12 $\pm$ 0.44 cm long and 0.72 $\pm$ 0.40 cm wide), and smaller than that of the *M. nitidum* clade (8.12 $\pm$ 2.47 cm long and 4.98 $\pm$ 1.69 cm wide). Additionally, as observed for thallus size, the cell size of species within the *G. brasiliensis* clade (9.06 $\pm$ 1.36 μm long and 6.06 $\pm$ 0.68 μm wide) was larger than that of the species in the *Monostroma* sp. clade and smaller than that of the species in the *M. nitidum* clade. Moreover, for the species within the *M. nitidum* clade, thallus thickness measured 33.80–34.20 μm, which was thicker than that of the *G. brasiliensis* and *Monostroma* sp. clades (Figure 4C,F,I,L,O and Table 2).

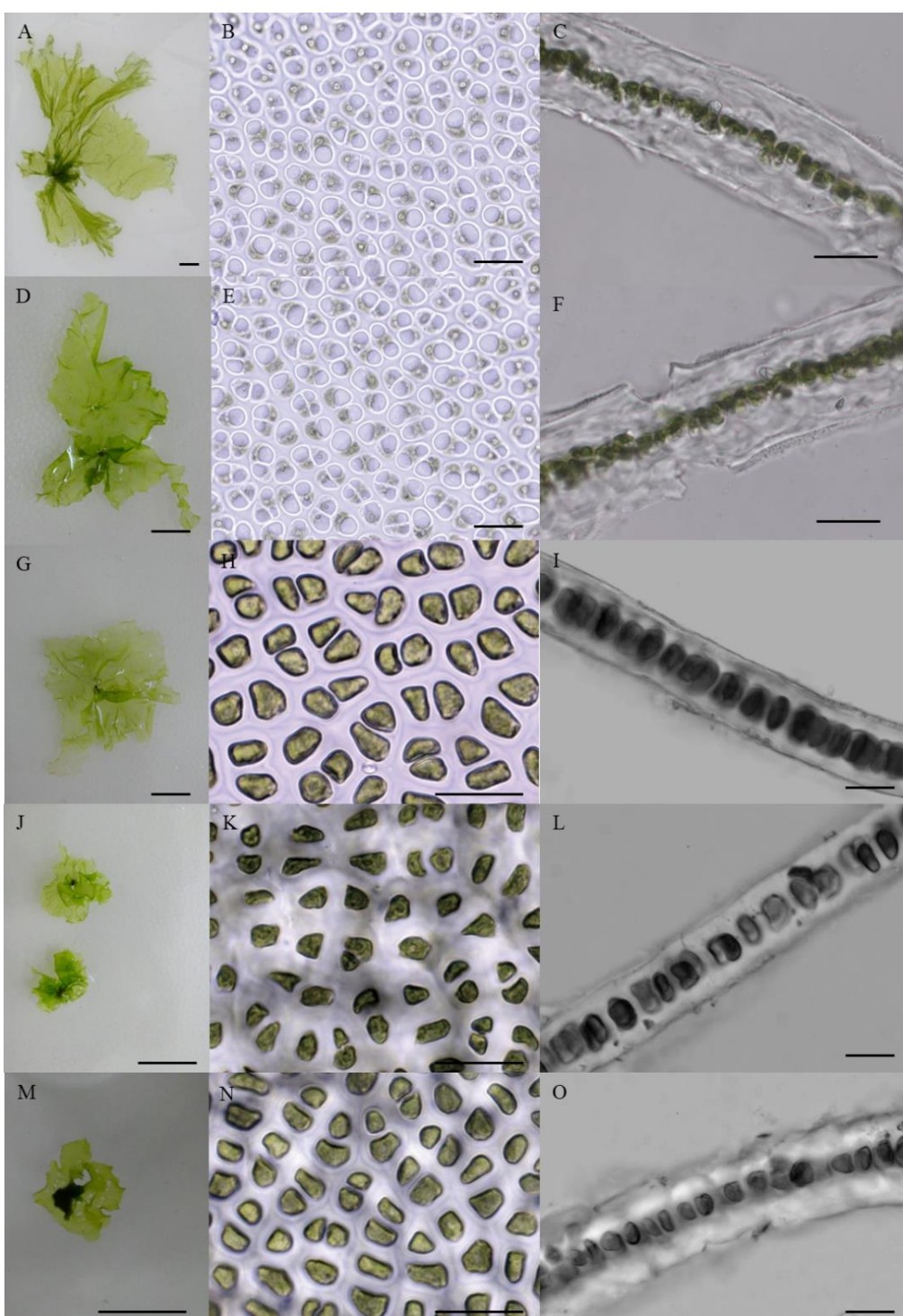

**Figure 4.** Morphology of the monostromatic green algae collected from the South China Sea. (**A**–**C**) YJ05 wild-living strain; (**D**–**F**) ST01 wild-living strain; (**G**–**I**) YJ02 wild-living strain; (**J**–**L**) MM01 wild-living strain; (**M**–**O**) ZH01 wild-living strain; (**A,D,G,J,M**) wild thallus of each strain; (**B,E,H,K,N**) cells of each strain in surface view; (**C,F,I,L,O**) cross-sectional views of each strain. Scale bars in the macro and microphotographs represent 1 cm and 20 μm, respectively.

**Table 2.** Thallus morphology of the monostromatic green algae collected from the South China Sea.

| Sample Name | Percentage of Cells with 1 to 2 Pyrenoids [a] | | Cell Size (Length × Width μm) [b] | Thallus Thickness (μm) [b] |
|---|---|---|---|---|
| | **1** | **2** | | |
| YJ02 | 96% | 4% | $9.06 \pm 1.36 \times 6.06 \pm 0.68$ | $25.2 \pm 1.9$ |
| YJ05 | 98% | 2% | $11.47 \pm 2.13 \times 8.42 \pm 1.72$ | $33.8 \pm 2.7$ |
| ST01 | 98% | 2% | $11.49 \pm 2.06 \times 8.44 \pm 1.52$ | $34.2 \pm 2.3$ |
| MM01 | 96% | 4% | $7.18 \pm 1.41 \times 5.47 \pm 1.04$ | $24.9 \pm 1.6$ |
| ZH01 | 96% | 4% | $7.15 \pm 1.28 \times 5.43 \pm 1.02$ | $26.3 \pm 2.5$ |

[a] Number of observed cells = 50. [b] Number of measured cells = 20, $X^- \pm s$.

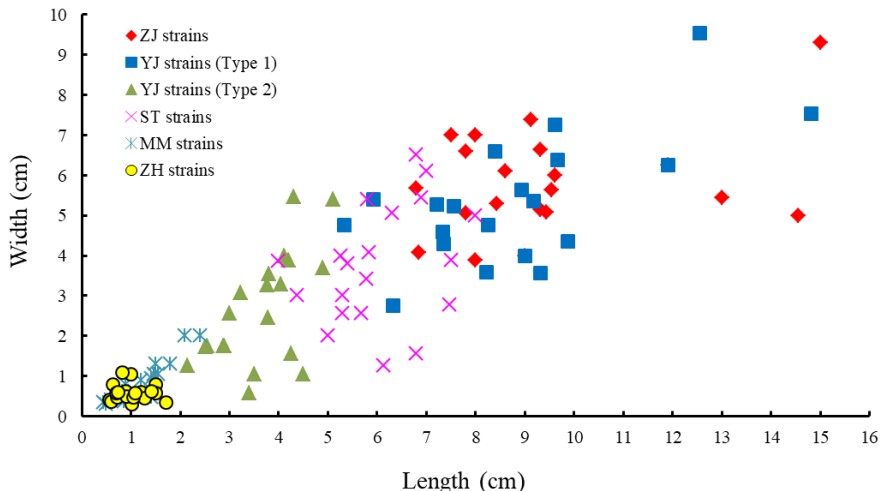

**Figure 5.** Thallus size of the monostromatic green algae collected from the South China Sea (n = 20). ZJ strains: Zhanjiang strains, including ZJ01-04 strains; YJ strains (Type 1): Yangjiang strains, including YJ04-05 strains; YJ strains (Type 2): Yangjiang strains, including YJ01-03 strains; ST strains: Shantou strains, including ST01-03 strains; MM strains: Maoming strains, including MM01-03 strains; ZH strains: Zhuhai strains, including the ZH01 strain.

## 4. Discussion

### 4.1. Taxonomic History of Monostromatic Green Algae

Monostromatic green algae, characterized by a single cell-layered blade-like thallus, were traditionally grouped under the eponymous genus *Monostroma*, which comprised *M. bullosum* (Roth) Wittrock and *M. oxyspermum* (Kützing) Doty [4,15]. Many species were subsequently added to this genus based on Thuret's description (Figure 6). For instance, *M. nitidum*, *M. arcticum*, *M. undulatum*, and others, were added to this genus by Wittrock (1866) [17]; *M. angicava* and *M. leptodermum* were added by Kjellman (1877) [39]; *M. groenlandicum* and *M. obscurum* were added by Agardh (1883) [40]; and *M. zostericola* was added by Tilden (1900) [41]. Kunieda (1934) erected the family Monostromaceae to accommodate all the related monostromatic green algal species [2]. However, as culture studies were conducted on their life cycle and thallus ontogeny, a number of taxonomic revisions on the genus *Monostroma* were proposed (Figure 6). Specifically, Gayral (1964) erected a new genus, *Ulvopsis*, with *U. grevillei* (synonymous to *M. grevillea* (Thuret) Wittrock) as a species type based on the sexual life cycle [42]. Bliding (1968) and Tatewaki (1969) recommended that *M. bullosum* Roth and *M. angicava* Kjellman should also be transferred to the genus *Ulvopsis* based on Gayral's identification [19,43]. Considering the shared life cycle patterns and anatomical features, Bliding also argued that taxa *M. leptoderma* and *M. zostericola* should be separated from the genus *Monostroma* and grouped under the newly erected *Kornmannia* genus [19]. Vinogradova (1969) proposed to include *M. groenlandicum* into the genus *Capsosiphon* based on shared morphology [3], *M. oxyspermum* into the newly

erected genus *Gayralia* (family Gayraliaceae) based on the typical asexual life cycle and the presence of a "tube" stage in the ontogeny, and *M. undulatum* into the newly erected genus *Protomonostroma* (Gayraliaceae) based on the absence of the above-mentioned "tube" stage in thallus ontogeny. After discovering the asexual life cycle of *M. latissimum*, Bast et al. (2009) questioned the taxonomic credibility of those groups defined based on life cycle type (i.e., sexual vs. asexual) and proposed to abolish the monotypic genus *Gayralia* and regroup *G. oxysperma* back to *Monostroma* as its original lectotype [44]. In contrast, Pellizzari et al. (2013) further increased the members of the genus *Gayralia* by naming a new species (*G. brasiliensis*) and also suggested adding *M. nitidum* from China and *M. kuroshiense* to this genus, based on morphology and molecular analysis [5]. In summary, there has been confusion and debate throughout the systematic revision history of monostromatic green algae, and it is necessary and important to clarify their current taxonomic status using effective and accepted taxonomic tools. For these reasons, the present study was designed, and it proposes to replace the genus *Gayralia* with the genus *Monostroma*.

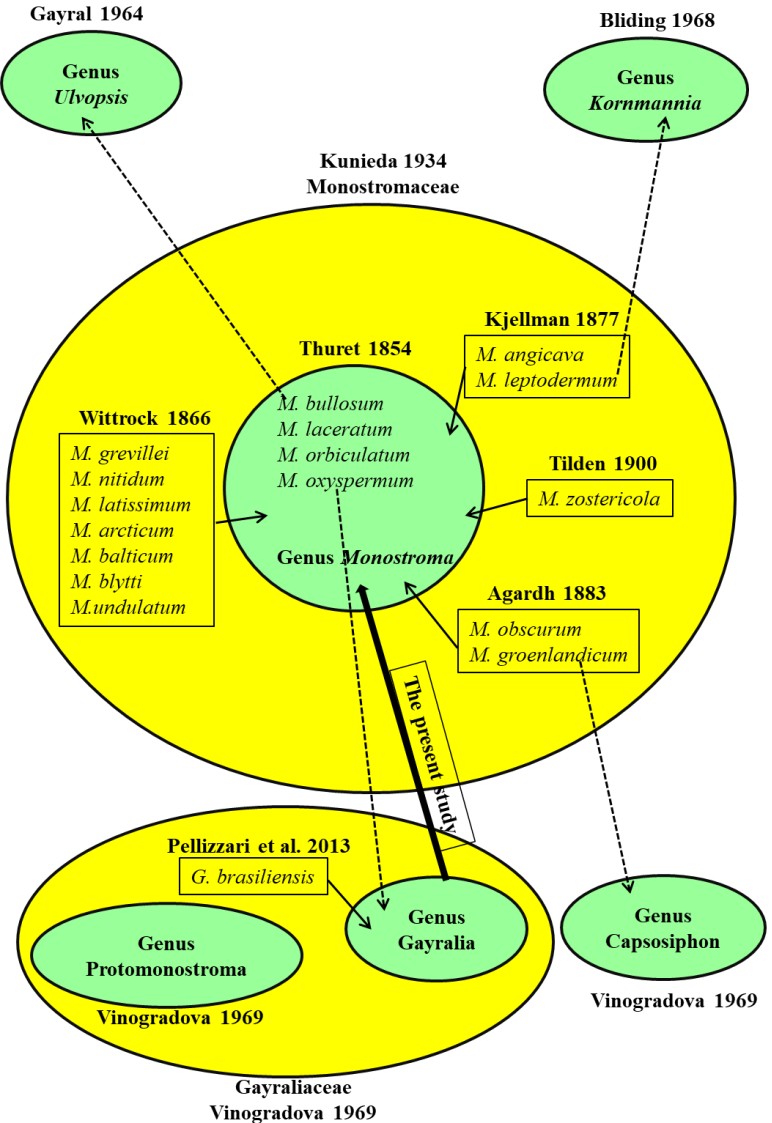

**Figure 6.** Taxonomic history of monostromatic green algae. Each taxon enclosed in the light green circles was regarded as a single genus, whereas each taxon enclosed in the yellow circles was regarded as a single family. The solid line arrows show the addition of species to the *Monostroma* genus; the dotted line arrows show the removal of species from the *Monostroma* genus; the bold line arrow represents the opinion of the present study [2,3,5,15,17,19,39–42].

*4.2. Taxonomic Assessment of Monostromatic Green Algae*

As with many other taxonomic groups of organisms, defining algal species is extremely difficult. Various concepts of the term species exist in systematics based on aspects that can be morphological, biological, genetic, ecological, paleontological, evolutionary, etc. [45]. It is well known that the concept of "morphological species" dominated algal systematics for many years, being based on "taximetrics", i.e., the overall similarity in morphology. Advancements in microscopy have enabled algal systematists to define new synapomorphies for the algal groups in question. Moreover, the morphology of the type specimens as a classification criterion has been widely used in algal systematics [10,21,22]. Based on the macro and microphotographs obtained in the present study, all the monostromatic green algae collected from the South China Sea showed a yellowish or light green color, were flat, and had a single-cell layer (Figure 4), which corresponds to their original morphological description by Thuret (1854) [15]. *Monostroma nitidum* was originally described by Wittrock (1866) as being monostromatic and thick, with a yellowish-green and shiny appearance and a frilly and mangled edge [17]. In the surface view, the cells are angular with slightly rounded corners and have an irregular arrangement, whereas in transverse sections, they are almost circular and 30–40 μm thick [17]. These morphological characteristics also match those of monostromatic species in the *M. nitidum* clade (Figure 4 and Table 2). In addition, the specimen type of *G. brasiliensis* is characterized by a single, expanded, laminar, monostromatic thallus. The fronds are ca. 10 cm broad, with a thickness of 25.0 ± 1.8 μm, and cell lumen of 9.0 ± 1.0 μm. In the surface view, cells are grouped into pairs, becoming more elongated toward the base. The cells are uninucleate with a large central vacuole, parietal chloroplast, and one or two pyrenoids [5]. These morphological traits are similar to those of monostromatic species in the *G. brasiliensis* clade, except for thallus size (Figure 4 and Table 2). Thus, the monostromatic green algae in the present study were classified into three different groups (*M. nitidum*, *G. brasiliensis*, and *Monostroma* spp) based on morphological analysis. Moreover, in our target samples, a similar morphology was monostromatic: thallus appearance, cell shape and arrangement, chloroplast position, and pyrenoid number. While the thallus shape, cell size, and thallus thickness of the samples presented distinct differences. These differences should be regarded as the typical morphological features of various *Monostroma* species. Our results were well supported by the studies of Bliding (1968) [19], Bast (2011) [4], Pellizzari et al. (2013) [5], and Cui et al. (2021) [10].

With the advent of DNA-based molecular barcoding technology, the concept of genetic species—which is based upon the genetic homogeneity of populations—has been increasingly taken into consideration, in addition to other concepts in algal systematics. DNA regions have been differentially used to construct phylogeny at different hierarchical levels, meaning that loci that evolve more slowly are used to analyze higher taxonomic levels, while those that evolve more rapidly are used to analyze relationships between closely related species [4,10,21,22]. ITS sequences were initially proposed for phylogeny reconstruction at or below the species level [46] due to the extensive sequence variation existing between members of closely related taxa. In this study, in line with the results of the morphological analysis, the data from ITS sequences also demonstrated that the monostromatic green algae were divided into three different clades (*M. nitidum* clade, *G. brasiliensis* clade, and *Monostroma* spp. clade) (Figure 2), which was further supported by the intraspecific genetic distance value (less than 0.6%, Figure 3). *Monostroma nitidum* (AF415170) from China, a reliable and accepted gene sequence, has been widely used as a reference to verify the identity of suspected *M. nitidum* specimens [4,10,47–49]. Thus, the monostromatic green algae (strain codes: ZJ01-04, YJ04-05, and ST01-03) were identified as *M. nitidum*. *Monostroma oxyspermum* was separated from the *Monostroma* genus due to its asexual life cycle and thallus ontogeny, and was subsequently included into the new *Gayralia* genus, currently comprising *G. brasiliensis* and *G. oxysperma* [3]. However, the taxonomic credibility of this species, defined based on sexuality, has been widely discredited [4,21,24]. In addition, asexual *G. oxysperma* have the same filament-sac-blade ontogeny as that of *M. nitidum* [43], while asexual *G. brasiliensis* have the same filament-

blade ontogeny as that of *M. latissimum* [5]. Moreover, this study surprisingly found that the genetic distance between *Monostroma* and *Gayralia* was lower than the interspecies distance within the *Monostroma* genus (Figure 3). Based on these observations, the present study agrees with the opinion of Bast (2011) and suggests that the genus *Gayralia* should be assigned to its original genus, *Monostroma*, and *G. brasiliensis* and *G. oxysperma* should therefore be renamed as *M. brasiliensis* and *M. oxyspermum*, respectively. Accordingly, the monostromatic green algae with the strain code YJ01-03 were identified as *M. brasiliensis*, and those with strain codes ZH01 and MM01-03 were identified as *Monostroma* spp.

## 5. Conclusions

In summary, the results of the present study demonstrated that the collected *Monostroma* and *Gayralia* specimens presented similar morphology in their single cell-layered thallus, with irregularly arranged cells, rounded cell corners, a parietal chloroplast, and predominantly one (>90%) pyrenoid. Furthermore, the inter-genera genetic distance between the *Monostroma* and *Gayralia* genera was lower than that observed within the *Monostroma* genus. Considering that the genus *Monostroma* was erected earlier than the genus *Gayralia*, it is here proposed to assign the genus *Gayralia* to the genus *Monostroma* based on the morphological and phylogenetic analysis and genetic distance data presented here.

**Author Contributions:** Conceptualization, J.C. and E.X.; methodology, C.C.; software, Y.H.; formal analysis, J.C., X.C. and J.L.; investigation, R.X. and B.H.; data curation, H.T.; writing—original draft preparation, J.C.; writing—review and editing, J.C., C.C. and E.X.; supervision, H.T. and E.X.; funding acquisition, J.C. All authors have read and agreed to the published version of the manuscript.

**Funding:** This research was funded by the PhD Start-up Foundation of Guangdong Ocean University, grant number R19049.

**Institutional Review Board Statement:** Not applicable.

**Data Availability Statement:** The datasets generated for this study can be found in the referenced online repositories. The genetic sequences amplified in this study are deposited in the GenBank repository and can be accessed through the accession numbers shown in Table 1 and Figure 5.

**Acknowledgments:** We would like to thank Dahai Gao for his helpful comments on the phylogenetic analysis.

**Conflicts of Interest:** The authors declare no conflict of interest.

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
