# Peer review of "Taxonomic Delimitation of the Monostromatic Green Algal Genera Monostroma Thuret 1854 and Gayralia Vinogradova 1969 (Ulotrichales, Chlorophyta)"

_diversity, doi:10.3390/d14090773_

Round 1

Reviewer 1 Report

See the attachment.

Reviewer 2 Report

The reviewed manuscript dedicated to delimitation of the genera Monostroma and Gayralia based on morphological and molecular-genetic analysis of numerous specimens. The results of the study are very important to modern taxonomy of green algae. An Introduction and Discussion provide sufficient background and include relevant references. The manuscript is well written and clear.

But I have some remarks to the authors.

Major comments:

1.      You wrote, that “We propose to assign genus Gayralia to genus Monostroma based on our morphology data, phylogenetic analysis and genetic distances”. But your arguments are not strong enough. I think it is necessary to give a detailed morphological description of the genera Monostroma and Gayralia (based on specimens of your study) and prove their morphological similarity. You have nice pictures on Figure 4, please, describe it in detail. Is it possible to analyze the secondary structure of ITS region? This would strengthen the results of phylogenetic analysis.

Minor comments:

Figure 6. Correct M.Obscurum to M.obscurum

Round 2

Reviewer 2 Report

The revised MS was corrected according to reviewer suggestions. I can recommend it for publication. 

Author Response

Response: We would like to extend our sincerest thanks to reviewer for constructive comments and suggestions on our manuscript. These comments and suggestions are greatly helpful to our continuing research.